# Effect of Japanese Cedar Pollen Sublingual Immunotherapy on Asthma Patients with Seasonal Allergic Rhinitis Caused by Japanese Cedar Pollen

**DOI:** 10.3390/biom12040518

**Published:** 2022-03-29

**Authors:** Shoko Ueda, Jun Ito, Norihiro Harada, Sonoko Harada, Hitoshi Sasano, Yuuki Sandhu, Yuki Tanabe, Sumiko Abe, Satomi Shiota, Yuzo Kodama, Tetsutaro Nagaoka, Fumihiko Makino, Asako Chiba, Hisaya Akiba, Ryo Atsuta, Sachiko Miyake, Kazuhisa Takahashi

**Affiliations:** 1Department of Respiratory Medicine, Faculty of Medicine and Graduate School of Medicine, Juntendo University, Tokyo 113-8431, Japan; sk-ueda@juntendo.ac.jp (S.U.); moisture@juntendo.ac.jp (J.I.); snharada@juntendo.ac.jp (S.H.); h-sasano@juntendo.ac.jp (H.S.); ysando@juntendo.ac.jp (Y.S.); yutanabe@juntendo.ac.jp (Y.T.); su-abe@juntendo.ac.jp (S.A.); sshiota@juntendo.ac.jp (S.S.); ykodama@juntendo.ac.jp (Y.K.); jnagaoka@juntendo.ac.jp (T.N.); makino@juntendo.ac.jp (F.M.); atsuta@juntendo.ac.jp (R.A.); kztakaha@juntendo.ac.jp (K.T.); 2Clinical Research Center, National Hospital Organization, Sagamihara National Hospital, Kanagawa 252-0392, Japan; 3Atopy (Allergy) Research Center, Graduate School of Medicine, Juntendo University, Tokyo 113-8421, Japan; 4Research Institute for Diseases of Old Ages, Graduate School of Medicine, Juntendo University, Tokyo 113-8421, Japan; 5Department of Immunology, Graduate School of Medicine, Juntendo University, Tokyo 113-8421, Japan; a-chiba@juntendo.ac.jp (A.C.); hisaya@juntendo.ac.jp (H.A.); s-miyake@juntendo.ac.jp (S.M.)

**Keywords:** Japanese cedar pollen, seasonal allergic rhinitis, asthma, sublingual immunotherapy, Japanese cedar pollinosis

## Abstract

Allergen immunotherapy is a promising treatment for allergic diseases that induce immune tolerance through the administration of specific allergens. In this study, we investigate the efficacy of sublingual immunotherapy (SLIT) in asthmatic patients with SAR-JCP and the dynamics of the parameters before and after treatment in a real-world setting. This was a prospective single-center observational study. Patients with asthma and SAR-JCP (*n* = 24) were recruited for this study and assessed using symptom questionnaires before SLIT and a year after the SLIT. In addition, a respiratory function test, forced oscillation technique, and blood sampling test were performed during the off-season before and after SLIT. The one-year SLIT for asthma patients with SAR-JCP significantly improved not only allergic rhinitis symptoms, but also asthma symptoms during the JCP dispersal season, and significantly improved airway resistance during the off-season. The change in the asthma control test and the visual analog scale score during the season before and after SLIT was negatively and positively correlated with the change in peripheral blood γδ T cells off-season before and after SLIT, respectively. It was suggested that improvement in asthma symptoms during the JCP dispersal season after SLIT was associated with reduced peripheral blood γδ T cells.

## 1. Introduction

Japanese cedar pollen (JCP) is widely scattered during spring from February to April. In Japan, perennial allergic rhinitis is more common in young people. However, after middle age, the prevalence of allergic rhinitis caused by JCP increases, and the prevalence of seasonal allergic rhinitis (SAR) caused by JCP (SAR-JCP) is 26.5% [1]. For bronchial asthma, nasal allergy is an independent risk factor for the exacerbation of asthma symptoms, the association of which is known as “one airway, one disease.” We previously reported the complication rate of pollinosis in patients with asthma in Japan, and a questionnaire survey using mobile phones showed a rate of approximately 60% [2]. According to various reports, 10–60% of patients with allergic rhinitis have asthma, and 28–85% of patients with asthma have allergic rhinitis [1,3,4,5,6]. JCP, which is one of the causative antigens of allergic rhinitis, has a large particle size of several tens of micrometers and does not reach the lower respiratory tract by itself. Therefore, it was previously thought to be unrelated to the exacerbation of asthma. However, it was recently found that a large number of fine particles, called orbicules, are attached to the surface of the JCP. Since the orbicules are approximately 1 μm, they can reach the airways and directly worsen asthma control. In Europe and North America, asthma has been reported to exacerbate frequently during the pollen dispersal season [7]. In Japan, Hojo et al. reported that asthma symptoms worsen during the JCP dispersal season, even asthma associated with SAR-JCP [8]. Respiratory dysfunction in SAR-JCP during JCP dispersal is thought to be caused by an increase in inflammatory mediators in the lower respiratory tract.

Since 2014, JCP sublingual immunotherapy (SLIT) has been covered by insurance as a new treatment for SAR-JCP, and its effectiveness and safety have been evaluated. In Europe and the United States, it has been reported that allergen immunotherapy (AIT) for birch pollinosis and grass pollinosis improves the symptoms of pollinosis, as well as asthma [9,10]. However, there is still little evidence on SLIT in Asia compared with Europe and the United States [11]. Similar to asthma associated with other types of SAR, treating asthma associated with SAR-JCP and JCP SLIT is expected to improve the symptoms of pollinosis as well as asthma. Kikkawa et al. reported that asthma exacerbation during the cedar pollen season in Japanese asthmatics with JCP can be sufficiently prevented by JCP SLIT [12]. However, these reports do not mention changes in airway resistance or cytokines.

Although the mechanisms contributing to asthma exacerbation by JCP remain unknown, several have been suggested, for example, direct exacerbation by orbicules, local release of mediators in the upper airways by nasal obstruction, systemic production of type 2 cytokines, activation of mast cells and/or basophils by IgE crosslinking, and subsequent activation of T cells. JCP SLIT is attracting attention as a treatment that radically improves asthma exacerbation caused by JCP.

Currently, JCP SLIT for allergic rhinitis and house dust mite (HDM) SLIT for allergic rhinitis are covered by insurance in Japan. HDM is a perennial allergen, but JCP is a seasonal allergen, and there is no evidence of the effect of SLIT on off-season asthma associated with SAR-JCP. Therefore, we investigated the effects of JCP SLIT on asthma concurrent with SAR-JCP.

## 2. Materials and Methods

### 2.1. Patients

This was a prospective single-center observational study. Patients with asthma associated with SAR-JCP provided written consent and were included in the study. January to April is the JCP scattering season in Japan. In this study, we described this period as in-season. We described June to October, which is a good time to start JCP SLIT, as the off-season (Appendix A). Patients who were aged 20 years or older and had asthma and SAR-JCP requiring JCP SLIT were recruited from our outpatient clinic at Juntendo University Hospital (Tokyo, Japan). The diagnosis of SAR-JCP was made by the criteria below, mainly based on rhinitis symptoms reported on the self-assessment of allergic rhinitis and asthma (SACRA) questionnaire and serum JCP-specific IgE antibodies as a reference to Japanese diagnostic criteria for allergic rhinitis: (1) in patients positive for ≥1 SACRA rhinitis symptom during the JCP dispersal season and JCP-specific IgE ≥ class 2, the diagnosis of SAR-JCP was confirmed; (2) in patients with no SACRA rhinitis symptoms during the JCP dispersal season or negative JCP-specific IgE (excluding patients taking antiallergic drugs), the diagnosis of SAR-JCP was excluded. Asthma was diagnosed based on a clinical history of episodic symptoms with airflow limitation and by variation in pulmonary function monitored by forced expiratory volume in 1 s or peak expiratory flow, according to Japanese guidelines [13]. Patients with any of the following criteria were excluded: (a) uncontrolled asthma and nonallergic asthma; (b) diagnoses of interstitial pneumonia, infectious disease, and cancer; (c) cases being treated with regular use of systemic corticosteroids; and (d) cases judged as inappropriate by the investigators. This study was reviewed and approved by Juntendo University Research Ethics Committee (Tokyo, Japan). Written informed consent was obtained from each patient before participation in the study. This study was conducted from January 2016 to October 2020 and registered in the UMIN Clinical Trial Registry (UMIN000020445) on 6 January 2016 (http://www.umin.ac.jp/, access on 27 November 2020).

### 2.2. Symptoms Scores

Subjective symptoms were evaluated using the asthma control test (ACT), the SACRA questionnaire, and medication score during the in-season (February to April) and off-season (June to October) before and after treatment. The contents of the ACT (Survey of Asthma Control in the Last 4 Weeks) and SACRA (Questionnaire and VAS Assessment of Rhinitis Symptoms and Asthma Symptoms) questionnaire have already been reported [14]. Medication scores for allergic rhinitis and asthma are assigned to the different medications as follows: 1 point = patient took nasal sprays and eye-drops; 2 points = patient took systemic antihistamines, inhaled corticosteroids, inhaled β2-agonists, inhaled anticholinergics, leukotriene receptor antagonist, suplatast tosilate, and theophylline; 3 points = patient took systemic corticosteroids, omalizumab, and mepolizumab; and score for the use of each drug will be multiplied by two in the case of use of maximum dose.

### 2.3. Measurement of Respiratory Function and Exhaled Nitric Oxide

Respiratory impedance was measured via the broadband frequency forced oscillation technique (FOT) using a commercially available device (MostGraph-01; Chest M.I. Co. Ltd., Tokyo, Japan) and met the standard recommendations. Spirometry was performed using a computed spirometer (Fukuda Denshi, Tokyo, Japan). The predicted values for forced expiratory volume in 1 s and vital capacity for the Japanese population were calculated using the formula proposed by the Japanese Respiratory Society. A fraction of exhaled nitric oxide (FeNO) tester (NIOX VERO, Aerocrine AB, Solna, Sweden) was used to measure the exhaled nitric oxide index.

### 2.4. Quantification of Circulating Lymphocyte Frequency

Flow cytometry analysis was conducted as previously described [15]. Briefly, peripheral venous blood samples were collected in heparin-containing tubes, and peripheral blood mononuclear cells (PBMCs; 3 × 10^6^/well) were purified by density-gradient centrifugation using Ficoll–Paque Plus solution (Cytiva, Tokyo, Japan). The cells were stained with different combinations of appropriate antibodies for 30 min at 4 °C. The following surface marker antibodies were used in this study: anti-CD3-APC-H7, anti-CD4-FITC, anti-CD19-FITC, anti-CD56-Alexa Fluor 700, anti-CD117 (c-Kit)-PE-CF594 (BD Biosciences, San Jose, CA, USA), anti-T-cell antigen receptor (TCR)-Pan-γδ-FITC, anti-TCR-Pan-γδ-PE (Beckman Coulter, Miami, FL, USA), anti-BDCA2-FITC, anti-CD1a-FITC, anti-CD11c-FITC, anti-CD14-FITC, anti-CD25-PE, anti-CD27-BV421, anti-CD34-FITC, anti-CD123-FITC, anti-CD127 (IL-7Rα)-BV421, anti-CD127-BV605, anti-CD161-PerCPCy5.5, anti-CD183 (CXCR3)-APC, anti-CD194 (CCR4)-BV510, anti-CD196 (CCR6)-PerCPCy5.5, anti-CD294 (CRTH2)-BV421, anti-FCɛR1-FITC, anti-Vα7.2-PE (BioLegend, San Diego, CA, USA), and anti-hCD1d tetramer loaded with PBS-57-APC (NIH tetramer core facility at Emory University). The negative lineage markers (Lin^−^) were CD1a^−^, CD3^−^, CD11c^−^, CD14^−^, CD19^−^, CD34^−^, TCRγδ^−^, CD123^−^, BDCA2^−^, and FCɛR1^−^. The Th1 cells were identified as CD3^+^, CD4^+^, CCR4^−^, CCR6^−^, and CXCR3^+^ cells. The Th2 cells were identified as CD3^+^, CD4^+^, CCR4^+^, CCR6^−^, and CXCR3^−^ cells. The Th17 cells were identified as CD3^+^, CD4^+^, CCR4^+^, CCR6^+^, and CXCR3^−^ cells. The regulatory T (Treg) cells were identified as CD3^+^, CD4^+^, CD25^+^, and CD127^−^ cells. The natural killer (NK) T (NKT) cells were identified as CD3^+^ and CD1d/PBS-57 tetramer^+^ cells. The γδ T cells were identified as CD3^+^ and TCRγδ^+^ cells. The mucosal-associated invariant T (MAIT) cells were identified as CD3^+^, Vα7.2 TCR^+^, and CD161 high cells. The NK cells were identified as CD3^−^ and CD56^+^ cells. Group 1 ILCs (ILC1s) were identified as Lin^−^, CD127^+^, CD161^+^, CD117^−^, and CRTH2^−^ cells. Group 2 ILCs (ILC2s) were identified as Lin^−^, CD127^+^, CD161^+^, and CRTH2^+^ cells. ILC3s were identified as Lin^−^, CD127^+^, CD161^+^, CD117^+^, and CRTH2^−^ cells. Dead cells were identified using the Zombie Fixable Viability Kit (BioLegend), followed by doublet exclusion of forward scatter and side scatter. After overnight fixation, the cells were analyzed using a fluorescence-activated cell sorting LSRFortessa cell analyzer (BD Biosciences). The fluorescence-activated cell sorting data were analyzed using FlowJo software (version 9; BD Biosciences).

### 2.5. Quantification of the Cytokine Concentration

The sera of the patients were collected using density-gradient centrifugation of blood samples and frozen at −80 ℃. The cytokine concentrations were measured using a multiplex assay, following the manufacturer’s instructions (Bio-Plex, Bio-Rad Laboratories, Hercules, CA, USA). The assay-working range was between the lower and upper limits of quantification (Appendix A). The serum interleukin (IL)-5, IL-10, IL-12, and VEGF concentrations were below the detection limit, and they were excluded from the analysis.

### 2.6. Statistical Analysis

Sample normality was examined using D’Agostino and Pearson’s tests. Differences in the parameters of the populations were analyzed for significance using the paired *t*-test, Mann–Whitney *U* test, Wilcoxon’s test, Fisher’s exact test, and chi-squared test, as needed. For the correlation between variables, Pearson’s and Spearman’s rank correlation coefficients were used, where appropriate. Differences were considered statistically significant when *p*-values were ≤0.05. Statistical analyses were performed using GraphPad Prism version 6 (GraphPad Software, San Diego, CA, USA).

## 3. Results

### 3.1. Participants and Background

Twenty-four patients were enrolled in this study. All patients received allergen-specific immunotherapy via the sublingual route in the form of sublingual immunotherapy liquid. Four patients had to discontinue the study because one patient developed skin itching as an adverse event, one patient became pregnant, and two patients were transferred to another hospital (Appendix A). There were no dropouts due to asthma exacerbation. Consequently, 20 patients were included for the entire study duration. Patient characteristics are shown in Table 1. The male-to-female ratio was 10:10, and the mean (± standard deviation) age was 50.9 ± 13.0 years. The mean duration of asthma was 13.8 ± 13.8 years. The counts of patients who were positive for each antigen-specific IgE were as follows: single positive for only JCP, 2 (10%); only JCP and Japanese cypress, 6 (30%); and multiple allergens, 12 (60%) (Table 1).

### 3.2. The Symptoms of SAR-JCP and Asthma

We compared the symptoms of SAR-JCP and asthma in-season before and after JCP SLIT using the SACRA and ACT questionnaires. According to the SACRA questionnaire, the subjective symptoms before SLIT during the in-season were runny nose (95%), sneezing (90%), nasal congestion (65%), nasal pruritus (80%), and ocular symptoms (95%). During the in-season, a year after the start of JCP SLIT, all subjective symptoms, such as a runny nose (40%), sneezing (25%), nasal congestion (10%), nasal pruritus (25%), and ocular symptoms (60%), were significantly reduced relative to the season before SLIT (Figure 1A). The pre-SLIT VAS value for rhinitis was 8.15 cm; however, it decreased to 2.68 cm post-SLIT (Figure 1B). The VAS score for asthma decreased from 5.05 cm pre-SLIT to 1.91 cm post-SLIT (Figure 1C). Based on the Japanese guidelines for the control level for SACRA, five patients (25%) had partially controlled asthma (1, 2 points), and nine patients (45%) had uncontrolled asthma (3, 4 points) before SLIT. After SLIT, the number of partially controlled patients decreased to three (15%), and the number of uncontrolled patients decreased to two (10%) (Figure 1D). Additionally, the pre-SLIT mean ACT score was 19 points, whereas the post-SLIT mean ACT score increased significantly to 22 points (*p* < 0.05) (Figure 1E). During the in-season, there was no difference in mean medication score between pre-SLIT and post-SLIT (7.4 ± 3.7 vs. 7.3 ± 3.4, *p* = 0.87).

### 3.3. Changes in Each Parameter after SLIT during the Off-Season

Based on the SACRA questionnaire responses, the subjective symptoms before SLIT during the off-season were runny nose (29.4%), sneezing (11.8%), nasal congestion (17.6%), nasal pruritus (23.5%), and ocular symptoms (23.5%) (Figure 1F). The VAS score for rhinitis was 0.71 cm during the same period (Figure 1B). These findings suggest that approximately one-fifth of the patients enrolled in this study had perennial hay fever. The symptoms of off-season hay fever did not change after the SLIT (Figure 1F). In addition, their asthma control was good, and there were no significant differences in the off-season asthma symptoms and assessment scores, including ACT scores, VAS scores, or asthma control status, before and after SLIT (Figure 1C–E and Table 2). Respiratory function and FeNO during the off-season did not change before and after SLIT (Figure 2A–E and Table 2). However, the airway resistance and reactance test using FOT showed that all parameters of resistance values at 5 Hz and 20 Hz, reactance at 5 Hz, resonant frequency, and low-frequency reactance area were significantly improved after SLIT (Figure 2F–J and Table 2). No significant differences were observed in the leukocyte fraction of peripheral blood and serum parameters, such as total IgE, IgG4, and thymus and activation-regulated chemokine concentrations, before and after SLIT, excluding decreased serum interferon-γ (IFN-γ) concentrations (Table 2). During the off-season, there was no difference in the mean medication score between pre-SLIT and post-SLIT (7.3 ± 2.7 vs. 6.3 ± 3.1, *p* = 0.14).

Next, we examined the frequency of peripheral blood PBMCs using flow cytometry. The gating strategy for PBMCs is shown in Appendix A. The frequencies of Th cells, ILCs, and MAIT cells were determined by their ratios to CD3^+^ and CD4^+^ cells, Lin^−^ CD127^+^ and CD161^+^ cells, and CD3^+^ cells, respectively. The frequencies of NK, NKT, and γδ T cells are also shown by their ratios to lymphocytes. The lymphocyte fraction during the off-season before and after SLIT, as measured by flow cytometry, showed a significant decrease in Treg cells and a significant increase in ILC3s (Table 2).

### 3.4. Association between Changes in SAR-JCP and Asthma Symptoms after SLIT during the In-Season and Changes in Each Parameter after SLIT during the Off-Season

Finally, we investigated whether improving the symptoms of both SAR-JCP and asthma during the JCP dispersal season was associated with changes in each of the biomarkers, including serum cytokines and lymphocyte fractions during the off-season before and after SLIT. The difference between the ACT scores before and after SLIT during the JCP dispersal season was positively correlated with the changes in the frequency of peripheral blood CD27-positive CD4^+^ T cells after SLIT during the off-season and negatively correlated with the changes in the frequency of peripheral blood CD27-negative CD4^+^ T cells and γδ T cells after SLIT during the off-season (Table 3 and Figure 3A–C). Similarly, the difference between the VAS scores for asthma during the season before and after SLIT was negatively correlated with the changes in the frequency of peripheral blood ILC3s during the off-season after SLIT and positively correlated with the changes in the frequency of peripheral blood ILC1s and γδ T cells during the off-season after SLIT (Table 3 and Figure 3D–F). In addition to the increase in ILC3s after SLIT, these findings suggest that the improvement of asthma symptoms during the JCP dispersal season after SLIT was associated with an increase in peripheral blood ILC3s and decrease in CD27-negative CD4^+^ and γδ T cells during the off-season.

## 4. Discussion

One year of JCP SLIT in asthma patients with SAR-JCP improved their SAR-JCP symptoms and asthma symptoms based on the SACRA questionnaire and ACT scores during the JCP dispersal season. JCP SLIT also significantly improved airway resistance, decreased serum IFN-γ concentrations and circulating Treg cells, and increased circulating ILC3s during the off-season. Moreover, the change in ACT scores during the season before and after SLIT was negatively correlated with the change in circulating CD27-negative CD4^+^ T cells and γδ T cells during the off-season before and after SLIT. The change in the VAS score for asthma during the season before and after SLIT was negatively correlated with the change in the frequency of circulating ILC3s and positively correlated with the change in the frequency of circulating ILC1s and γδ T cells during the off-season before and after SLIT. To our knowledge, this is the first study to suggest that improvements in asthma symptoms during the JCP dispersal season after SLIT are associated with a decrease in the frequency of circulating CD27-negative CD4^+^ T cells and γδ T cells and an increase in the frequency of circulating ILC3s during the off-season before and after SLIT.

JCP SLIT improved rhinitis and asthma symptoms in this study, and some previous reports have suggested that treatment for rhinitis, including intranasal corticosteroid, antihistamines, and SLIT, improve asthma symptoms [12,16,17,18,19]. SLIT was effective for some allergic asthma and asthma exacerbations during the JCP dispersal season in Japanese patients with asthma, and SAR-JCP could be sufficiently prevented by JCP SLIT [12,20]. A study of patients with mild-to-moderate asthma coexisting with allergic rhinitis showed a control of symptoms for 16 weeks after intranasal corticosteroid therapy alone, despite the absence of inhaled corticosteroid [17]. The relationship between allergic rhinitis and asthma can be explained by the concept of “one airway, one disease.” Both upper and lower respiratory tract diseases coexist and share similar pathophysiological and inflammatory profiles [6,21]. It has been reported that direct stimulation by antigens caused by both the upper and lower respiratory tract infection or inflammation of the upper and lower respiratory tracts were linked [19,22,23,24,25]. Endoscopy after the nasal cavity or bronchus was directly stimulated with an antigen showed increased expressions of inflammatory mediators in the nose after stimulating the bronchi and in the bronchi after stimulating the nose [19,22,23,24,25]. These indicate that antigen challenge in one airway (nasal or bronchial) induces allergic inflammation in the other airway. It has been suggested that local inflammation induces inflammation at different sites, probably through the hematogenous dissemination of inflammatory mediators, inflammatory cells, or immunocompetent cells. However, this has not been established.

In contrast to the concept of “one airway, one disease,” a meta-analysis including trials involving 477 patients with asthma outcomes failed to show a statistically significant benefit of intranasal corticosteroid therapy in asthma [26]. In the present study, comparisons of off-season SAR-JCP and asthma symptoms showed no significant difference after SLIT. Furthermore, it was assumed from the concept of “one airway, one disease” that the symptoms of rhinitis and asthma were associated. However, Table 3 demonstrated no association between rhinitis and asthma symptoms. It has previously been reported that asthma patients with hay fever have worse symptoms of asthma during the pollen dispersal season than during the off-season. However, in our present study, the symptoms during the off-season were originally minor, which is why there were no differences in off-season symptoms [7,8].

Of the participants in this study, 10–35% had conjunctivitis and rhinitis even during the off-season, which was probably caused by antigens other than cedar pollen, such as house dust mites, molds, and gramineous pollens. The results of this study indicate that JCP SLIT does not affect conjunctivitis or rhinitis symptoms other than those related to SAR-JCP. For patients with conjunctivitis or rhinitis symptoms persisting during the off-season, it is necessary to separately consider the AIT for each allergy.

It has been reported that SLIT improved respiratory function one year later [27]. In the present study, the FOT parameters, including airway resistance, improved; however, no significant differences in respiratory function and FeNO were observed during the off-season. The reasons for these are that this study included patients who had stable off-season asthma control and those who had been stepping down asthma treatment because the research plan did not limit the change of treatments for asthma. A previous report showed that HDM SLIT improved airway wall thickening in patients with asthma [28]. Similar to HDM SLIT, JCP SLIT may have improved airway wall thickening, which may have improved airway resistance. The FOT is a method for measuring airway resistance, which has been clinically applied in recent years. This study suggested that changes that could not be detected by spirometry could be assessed using the FOT.

It has been envisioned that AIT induces immunomodulation of the allergic immune responses. During immunomodulation, changes in allergen-specific memory T- and B-cell responses decrease IgE and increase IgG4 from B cells, downregulate mast cell and basophil activation, and suppress Th2 cell proliferation and the production of relevant Th2 cytokines, such as IL-4, IL-5, and IL-13, all of which lead to the suppression of allergic symptoms [29,30,31,32]. AIT has also been shown to inhibit the seasonal increases in the concentrations of circulating ILC2s, upregulate the activation of allergen-specific Treg subsets, and downregulate dysfunctional allergen-specific Treg subsets [33,34]. The changes in serum cytokines and lymphocyte fractions after SLIT, including reductions in IFNγ and Treg and an increase in ILC3s, in this study were partially different from those previously reported. The reasons for these discrepancies were envisioned, as there was a difference between our study, which targeted blood sampling during the off-season, and previous studies, which targeted the reaction of the perennial antigen or was conducted during the pollen dispersal season [29,33,34,35,36]. No reaction to ILC2s was observed during the off-season [33]. The decrease in Tregs may have resulted from a decrease in the dysfunctional allergen-specific Treg subsets and their diminished need due to the subsided inflammation associated with rhinitis and asthma. Furthermore, this study showed that an increase in the frequency of ILC3s and a decrease in γδ T cells and CD27-negative CD4^+^T cells during the off-season were associated with an improvement in asthma symptoms in patients with SAR-JCP during the in-season. Our mouse studies showed that CD27-negative CD4^+^ T cells produce Th2 cytokines, such as IL-4, IL-5, and IL-13, and a previous report on patients with alder pollen allergy showed that CD27-negative allergen-specific CD4^+^ T cells were reduced by allergen-specific immunotherapy [37,38]. γδ T cells are also involved in asthma pathogenesis and Th2 inflammation through IL-17 production, and this trend is considered to be similar to that of CD27-negative CD4^+^ T cells [39,40]. ILC3s are known to be involved in obesity, asthma, and IL-17 production in mouse studies, but an association between the increase in circulating ILC3s and the improvement of asthma symptoms in our study was not established [41,42]. No significant changes in ILC2s associated with type 2 inflammation were observed after SLIT, and further investigations of the associations between the changes in lymphocyte fractions, the improvement of asthma symptoms, and the absence of an association with rhinitis signs are needed.

The present study had several limitations. First, almost all participants in this study were allergic to allergens other than JCP, which may have influenced the off-season measurement trends owing to possible complementary contributions of allergic inflammation progressing to asthma and SAR. Second, the asthma control of the participants in this study was not very poor before SLIT. It may have been difficult to see a significant difference in the short-term evaluation of one year. Third, this study started during the off-season, ended the next off-season, and did not carry out investigations, including blood sampling and respiratory function tests, during the in-season before SLIT and a year after SLIT. AIT is more effective after a few years and requires a longer study period. Fourth, this study included a few patients who were recruited from a single medical institution. However, this allowed the studying of a relatively homogeneous patient population receiving essentially identical levels of treatment from specialists and an investigation based on detailed information unavailable in most epidemiological surveys. Therefore, further studies with larger cohorts are required to verify these results.

## 5. Conclusions

SLIT for asthma patients with SAR-JCP resulted in the improvement of asthma and SAR-JCP symptoms in-season. The findings suggest that SLIT for asthma patients with SAR-JCP contributed to the further stabilization of asthma symptoms, and changes in peripheral blood CD27-negative CD4^+^ T cells, γδ T cells, ILC1s, and ILC3s during the off-season were associated with improved asthma symptoms after SLIT.

## Figures and Tables

**Figure 1 biomolecules-12-00518-f001:**
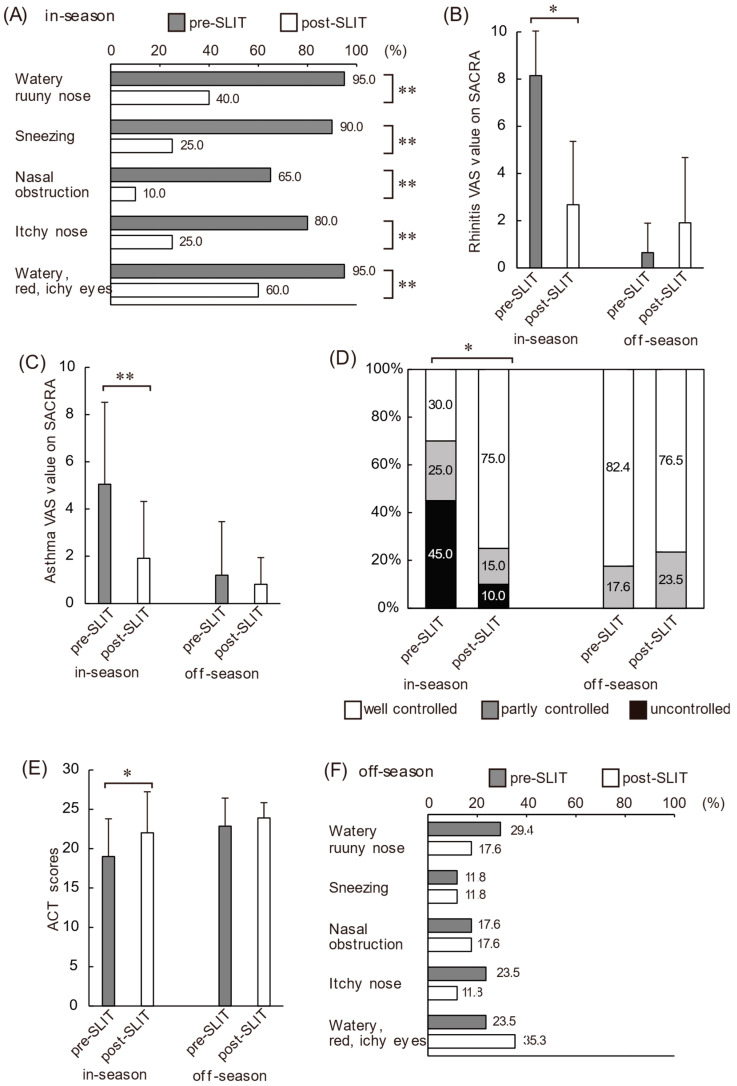
Changes in ACT and SACRA questionnaire scores during the in-season (February to April) and off-season (June to October) before and after SLIT. (**A**) Percentage of patients answering ‘yes’ to each question of the SACRA questionnaire about rhinitis symptoms before and after SLIT in-season (*n* = 20). Rhinitis (**B**) and asthma (**C**) control levels measured by VAS before and after SLIT. (**D**) Proportion of patients with asthma control based on SACRA surveys. (**E**) Asthma control level measured by ACT before and after SLIT. (**F**) Percentage of patients answering ‘yes’ to each question of the SACRA questionnaire about rhinitis symptoms before and after SLIT during the off-season (*n* = 17). Data are presented as the mean ± SD. * *p* < 0.05, ** *p* < 0.001. Abbreviations: ACT, asthma control test; SACRA, self-assessment of allergic rhinitis and asthma.

**Figure 2 biomolecules-12-00518-f002:**
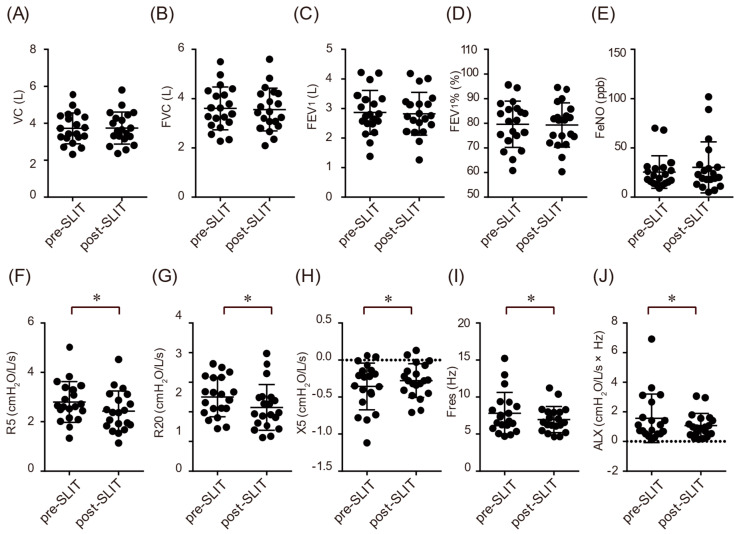
Changes in respiratory function (VC (**A**), FVC (**B**), FEV_1_ (**C**), FEV_1_% (**D**)), FeNO levels (**E**), and forced oscillation technique (FOT) parameters (R5 (**F**), R20 (**G**), X5 (**H**), Fres (**I**), ALX (**J**)) before and after SLIT during the off-season. The values of FOT parameters are presented in whole breath. Bars indicate median values. Abbreviations: ALX, a low-frequency reactance area; FeNO, a fraction of exhaled nitric oxide; FEV_1_, forced expiratory volume in one second; Fres, resonant frequency; FVC, forced vital capacity; R5 and R20, resistance at 5 Hz and 20 Hz; X5, reactance at 5 Hz; VC, vital capacity. The measurements were averaged over several tidal breaths (whole-breath analysis). Each bar represents the mean ± SD. * *p* < 0.05.

**Figure 3 biomolecules-12-00518-f003:**
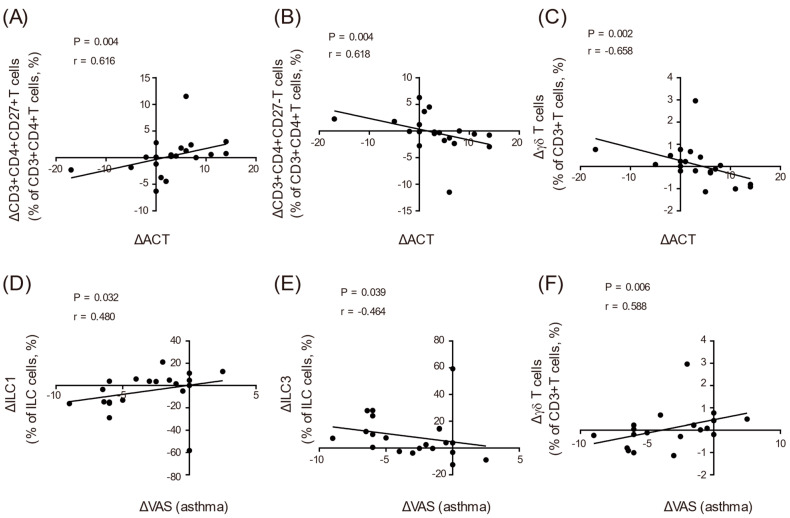
The change in asthma symptoms (ACT score (**A**–**C**) and VAS value (**D**–**F**)) after SLIT during the in-season was associated with the change in the percentage of circulating CD27-positive CD4^+^ T cells (**A**), CD27-negative CD4^+^ T cells (**B**), γδ T cells (**C**,**F**), ILC1s (**D**), and ILC3s (**E**) in the off-season before and after SLIT. Abbreviations: ACT, asthma control test; ILC1, group 1 innate lymphoid cell; ILC3, group 3 innate lymphoid cell; SACRA, self-assessment of allergic rhinitis and asthma.

**Table 1 biomolecules-12-00518-t001:** Baseline characteristics of the study population.

*n* = 20	Number (%) or Mean ± SD
Male, *n* (%)	10 (50.0%)
Female, *n* (%)	10 (50.0%)
Age (y), mean (±SD)	50.9 ± 13.0
Age at asthma onset (y), mean (±SD)	37.1 ± 18.6
Duration of asthma (y), mean (±SD)	13.8 ± 13.8
BMI (kg/m^2^), mean (±SD)	24.1 ± 3.9
Smoking history	
Never-smoker, *n* (%)	17 (85.0%)
Ex-smoker, *n* (%)	3 (15.0%)
AERD, *n* (%)	0 (0%)
Atopic dermatitis, *n* (%)	5 (25.0%)
Atopic conjunctivitis, *n* (%)	2 (10.0%)
Chronic sinusitis, *n* (%)	2 (10.0%)
Childhood asthma, *n* (%)	4 (20.0%)
Hypertension, *n* (%)	3 (15.0%)
Hyperlipidemia, *n* (%)	2 (10.0%)
JGL treatment step	
Step 1, *n* (%)	2 (10.0%)
Step 2, *n* (%)	4 (20.0%)
Step 3, *n* (%)	12 (60.0%)
Step 4, *n* (%)	2 (10.0%)
Daily dose of ICS (FP equivalent dose, µg)	547.5 ± 302.0
Short-course oral corticosteroid therapy, *n* (%)	1 (5.0%)
Omalizumab therapy, *n* (%)	1 (5.0%)
Mepolizumab therapy, *n* (%)	1 (5.0%)
Antigen-specific IgE antibodies positive	
JCP, *n* (%)	20 (100.0%)
Grass pollen except for JCP, *n* (%)	18 (90.0%)
Dust mites, *n* (%)	11 (55.0%)
Mold, *n* (%)	1 (5.0%)
Animal dander, *n* (%)	4 (20.0%)

Abbreviations: AERD, aspirin-exacerbated respiratory disease; BMI, body mass index; FP, fluticasone propionate; ICS, inhaled corticosteroid; IgE, immunoglobulin E; JCP, Japanese cedar pollen; JGL, Japanese guideline; SD, standard deviation.

**Table 2 biomolecules-12-00518-t002:** Baseline characteristics of the study population and dynamics of the parameters in patients treated with SLIT.

	Pre-SLIT (*n* = 20)	Post-SLIT (*n* = 20)	*p*-Value
VAS score points (rhinitis symptoms), *n* = 17	0.7 ± 1.3	1.9 ± 2.8	0.109
SACRA questionnaire about rhinitis symptom, *n* = 17			
Runny nose	5 (29.4%)	3 (17.6%)	0.688
Sneezing	2 (11.8%)	2 (11.8%)	1.000
Nasal congestion	3 (17.6%)	3 (17.6%)	1.000
Pruritus nasal	4 (23.5%)	2 (11.8%)	0.656
Ocular symptoms	4 (23.5%)	6 (35.3%)	0.708
ACT score points	22.9 ± 3.6	23.9 ± 1.9	0.344
VAS score points (asthma symptoms), *n* = 17	1.0 ± 1.9	0.8 ± 1.1	0.930
SACRA questionnaire about asthma symptom, *n* = 17			
Well controlled/partly controlled/uncontrolled, *n* (%)	14 (82.4%)/3 (17.6%)/0 (0%)	13 (76.5%)/4 (23.5%)/0 (0%)	1.000
FeNO (ppb)	25.5 ± 16.5	30.3 ± 25.9	0.347
VC (L)	3.7 ± 0.8	3.7 ± 0.9	0.541
FVC (L)	3.6 ± 0.9	3.6 ± 0.9	0.205
FEV_1_ (L)	2.9 ± 0.7	2.8 ± 0.7	0.209
FEV_1_% (%)	79.6 ± 9.4	79.3 ± 9.0	0.720
R5 (cmH_2_O/L/s)	2.8 ± 0.8	2.4 ± 0.8	0.009 *
R20 (cmH_2_O/L/s)	2.5 ± 0.7	2.1 ± 0.8	0.002 *
X5 (cmH_2_O/L/s)	−0.4 ± 0.3	-0.3 ± 0.2	0.049 *
Fres (Hz)	7.8 ± 2.8	7.0 ± 1.8	0.039 *
ALX (cmH_2_O/L/s × Hz)	1.6 ± 1.6	1.1 ± 0.8	0.019 *
Peripheral neutrophils (%)	58.6 ± 9.8	58.5 ± 8.2	0.927
Peripheral neutrophils (cells/μL)	3506.0 ± 1037.0	3576.0 ± 1172.0	0.619
Peripheral eosinophils (%)	3.2 ± 2.0	3.5 ± 2.5	0.658
Peripheral eosinophils (cells/μL)	220.1 ± 171.2	210.1 ± 164.2	0.787
Peripheral basophils (%)	0.7 ± 0.6	0.7 ± 0.5	0.918
Peripheral basophils (cells/μL)	37.4 ± 31.3	38.4 ± 27.1	0.784
Peripheral lymphocytes (%)	31.7 ± 8.7	31.9 ± 7.6	0.863
Peripheral lymphocytes (cells/μL)	1839.0 ± 507.6	1914.0 ± 652.5	0.286
Peripheral monocytes (%)	5.3 ± 0.6	5.5 ± 1.3	0.877
Peripheral monocytes (cells/μL)	317.5 ± 80.9	325.0 ± 93.8	0.596
Total IgE (IU/mL), *n* = 19	313.3 ± 569.2	254.1 ± 276.6	0.761
IgG4 (mg/dL), *n* = 18	98.6 ± 174.9	89.9 ± 114.4	0.074
TARC (pg/mL), *n* = 6	301.3 ± 222.6	270.0 ± 145.3	0.813
IL-1β (pg/mL), *n* = 19	3.3 ± 5.4	2.6 ± 2.8	0.129
IL-1Rα (pg/mL), *n* = 18	327.8 ± 514.1	249.7 ± 284.9	0.159
IL-4 (pg/mL)	2.8 ± 1.2	2.6 ± 1.1	0.782
IL-7 (pg/mL), *n* = 15	11.4 ± 6.5	12.2 ± 6.2	0.484
IL-8 (pg/mL), *n* = 16	11.5 ± 4.8	11.1 ± 5.2	0.930
IL-9 (pg/mL), *n* = 19	81.8 ± 70.7	104.6 ± 73.0	0.575
IL-13 (pg/mL), *n* = 8	6.7 ± 2.6	6.0 ± 1.6	0.482
IL-17 (pg/mL), *n* = 19	30.3 ± 15.9	26.2 ± 13.7	0.112
Eotaxin-1 (pg/mL)	92.7 ± 46.8	88.8 ± 42.7	0.648
Basic FGF (pg/mL), *n* = 18	30.2 ± 10.9	33.5 ± 11.5	0.404
G-CSF (pg/mL), *n* = 19	45.9 ± 20.9	49.5 ± 19.0	0.275
IFN-γ (pg/mL), *n* = 19	57.8 ± 79.5	38.6 ± 49.9	0.007 *
IP-10 (pg/mL)	614.1 ± 259.9	575.0 ± 203.5	0.425
MCP-1 (pg/mL), *n* = 12	28.9 ± 40.3	26.0 ± 19.7	0.176
MIP-1α (pg/mL), *n* = 19	3.1 ± 1.8	2.7 ± 1.6	0.166
PDGF-BB (pg/mL)	1805.0 ± 994.8	1617.0 ± 898.3	0.409
MIP-1β (pg/mL)	145.8 ± 66.3	169.1 ± 61.5	0.154
RANTES (pg/mL)	6714.0 ± 1025.0	6998.0 ± 981.8	0.294
TNF-α (pg/mL), *n* = 19	103.9 ± 85.4	100.8 ± 39.0	0.832
Th1 cells (% of Th cells, %)	19.4 ± 6.1	19.8 ± 8.6	0.697
Th2 cells (% of Th cells, %)	6.1 ± 2.7	6.3 ± 2.9	0.672
Th17 cells (% of Th cells, %)	6.0 ± 1.4	5.6 ± 1.8	0.232
Treg cells (% of Th cells, %)	7.4 ± 3.5	6.1 ± 3.8	0.001 *
ILC1 (% of ILC cells, %)	61.7 ± 15.1	56.7 ± 18.6	0.222
ILC2 (% of ILC cells, %)	25.0 ± 12.9	22.0 ± 12.4	0.169
ILC3 (% of ILC cells, %)	12.2 ± 9.7	20.8 ± 15.7	0.030 *
NK cells (% of lymphoid cells, %)	14.7 ± 7.5	15.5 ± 7.6	0.522
γδ T cells (% of lymphoid cells, %)	2.4 ± 1.6	2.5 ± 1.6	0.978
NKT (% of lymphoid cells, %)	0.029 ± 0.030	0.028 ± 0.036	0.546
MAIT cells (% of CD3^+^cells, %)	3.4 ± 2.3	3.0 ± 1.6	0.312
CD27^+^CD4^+^ T cells (% of CD3^+^CD4^+^ T cells, %)	90.5 ± 6.4	90.8 ± 6.7	0.730
CD27^-^CD4^+^ T cells (% of CD3^+^CD4^+^ T cells, %)	9.5 ± 6.4	9.2 ± 6.7	0.736

Data are presented as the mean ± standard deviation unless otherwise indicated. * *p* < 0.05. Abbreviations: ACT, asthma control test; FeNO, fractional exhaled nitric oxide; FEV_1_, forced expiratory volume in one second; FEV_1_%, forced expiratory volume % in one second; FGF, fibroblast growth factor; FVC, forced vital capacity; G-CSF, granulocyte-colony stimulating factor; IFN-γ, interferon-gamma; IL, interleukin; ILC, innate lymphoid cell; IP, interferon-γ inducible protein; MAIT, mucosal-associated invariant T; MCP, monocyte chemotactic protein; MIP, macrophage inflammatory protein; NK, natural killer; PDGF, platelet-derived growth factor; RANTES, regulated on activation normal T cell expressed and secreted; TARC, thymus and activation-regulated chemokine; Th, helper T; TNF-α, tumor necrosis factor-α; Treg, regulatory T; VAS, visual analog scale; VC, vital capacity.

**Table 3 biomolecules-12-00518-t003:** Association between patient symptoms during the in-season and each parameter during the off-season.

	ΔACT		ΔVAS (Asthma Symptoms)		ΔVAS (Rhinitis Symptoms)	
	r	*p*-Value	r	*p*-Value	r	*p*-Value
ΔACT			−0.785	<0.001 *	−0.295	0.207
ΔVAS (asthma symptoms)	−0.785	<0.001 *			0.075	0.753
ΔVAS (rhinitis symptoms)	−0.295	0.207	0.075	0.753		
ΔFeNO (ppb)	0.113	0.636	−0.062	0.794	−0.081	0.735
ΔVC (L)	−0.201	0.396	0.316	0.174	−0.229	0.331
ΔFVC (L)	−0.088	0.713	0.048	0.841	−0.149	0.530
ΔFEV1 (L)	0.050	0.835	−0.232	0.324	0.033	0.890
ΔFEV1% (%)	0.205	0.385	-0.391	0.088	0.246	0.295
ΔR5 (cmH_2_O/L/s)	−0.286	0.221	0.057	0.812	−0.127	0.593
ΔR20 (cmH_2_O/L/s)	−0.067	0.781	−0.101	0.673	−0.126	0.597
ΔX5 (cmH_2_O/L/s)	−0.058	0.809	0.018	0.940	−0.108	0.650
ΔFres (Hz)	0.035	0.884	0.019	0.936	−0.037	0.878
ΔALX (cmH_2_O/L/s × Hz)	0.040	0.867	−0.160	0.501	0.178	0.453
ΔPeripheral neutrophils (%)	0.110	0.643	−0.276	0.239	0.078	0.744
ΔPeripheral neutrophils (cells/μL)	−0.070	0.771	0.013	0.958	0.157	0.509
ΔPeripheral eosinophils (%)	0.153	0.519	−0.238	0.312	0.033	0.891
ΔPeripheral eosinophils (cells/μL)	0.148	0.533	−0.048	0.840	−0.285	0.223
ΔPeripheral bosophils (%)	0.075	0.755	0.150	0.527	0.048	0.840
ΔPeripheral basophils (cells/μL)	0.082	0.730	0.019	0.936	−0.263	0.263
ΔPeripheral lymphocytes (%)	−0.157	0.510	0.235	0.318	0.134	0.573
ΔPeripheral lymphocytes (cells/μL)	−0.187	0.429	0.380	0.098	0.143	0.547
ΔPeripheral monocytes (%)	−0.167	0.482	0.235	0.318	−0.079	0.739
ΔPeripheral monoocytes (cells/μL)	−0.222	0.347	0.328	0.157	−0.022	0.926
ΔTotal IgE (IU/mL), *n* = 19	−0.245	0.313	0.240	0.322	−0.317	0.186
ΔIgG4 (mg/dL), *n* = 18	0.059	0.815	−0.044	0.863	−0.320	0.195
ΔTARC (pg/mL), *n* = 6	−0.273	0.617	−0.203	0.700	−0.371	0.497
ΔIL-1β (pg/mL), *n* = 19	0.072	0.768	0.128	0.601	−0.358	0.132
ΔIL-1Rα (pg/mL), *n* = 18	0.011	0.964	0.215	0.392	−0.415	0.087
ΔIL-4 (pg/mL)	−0.275	0.241	0.357	0.122	0.121	0.611
ΔIL-7 (pg/mL), *n* = 15	−0.171	0.539	−0.016	0.956	0.109	0.697
ΔIL-8 (pg/mL), *n* = 16	−0.334	0.204	0.229	0.389	−0.210	0.431
ΔIL-13 (pg/mL), *n* = 8	−0.238	0.582	0.878	0.008 *	−0.771	0.033 *
ΔIL-17 (pg/mL), *n* = 19	−0.313	0.191	0.438	0.061	−0.202	0.407
ΔEotaxin (pg/mL)	−0.166	0.484	0.223	0.344	0.074	0.758
ΔBasic-FGF (pg/mL), *n* = 18	−0.276	0.268	−0.203	0.419	0.083	0.744
ΔG-CSF (pg/mL), *n* = 19	−0.297	0.216	0.198	0.416	−0.177	0.468
ΔIFN-γ (pg/mL), *n* = 19	−0.107	0.664	0.260	0.282	−0.631	0.004 *
ΔIP-10 (pg/mL)	0.086	0.718	0.029	0.904	−0.019	0.937
ΔMCP-1 (pg/mL), *n* = 12	−0.338	0.280	0.226	0.477	−0.187	0.558
ΔMIP-1α (pg/mL), *n* = 19	−0.282	0.242	0.309	0.198	−0.367	0.122
ΔPDGF-BB (pg/mL)	−0.134	0.574	0.387	0.092	−0.063	0.791
ΔMIP-1β (pg/mL)	−0.355	0.124	−0.102	0.668	0.465	0.039 *
ΔRANTES (pg/mL)	−0.302	0.195	0.419	0.066	0.003	0.990
ΔTNF-α (pg/mL), *n* = 19	0.140	0.567	−0.296	0.219	−0.289	0.230
ΔTh1 cells (% of Th cells, %)	−0.387	0.092	0.074	0.758	0.294	0.209
ΔTh2 cells (% of Th cells, %)	−0.257	0.274	−0.030	0.899	0.186	0.434
ΔTh17 cells (% of Th cells, %)	−0.247	0.294	0.135	0.572	−0.221	0.350
ΔTreg cells (% of Th cells, %)	−0.153	0.521	0.185	0.435	−0.045	0.850
ΔILC1 (% of ILC cells, %)	−0.372	0.107	0.480	0.032 *	−0.073	0.759
ΔILC2 (% of ILC cells, %)	−0.074	0.756	−0.157	0.508	0.293	0.209
ΔILC3 (% of ILC cells, %)	0.400	0.080	−0.464	0.039 *	−0.201	0.395
ΔNK cells (% of lymphoid cells, %)	−0.375	0.104	0.316	0.174	0.182	0.441
Δγδ T cells (% of lymphoid cells, %)	−0.658	0.002 *	0.588	0.006 *	0.269	0.251
ΔNKT (% of lymphoid cells, %),	0.177	0.456	0.128	0.590	0.075	0.754
ΔMAIT cells (% of CD3+cells, %)	−0.010	0.967	−0.162	0.496	0.203	0.391
ΔCD27^+^CD4^+^ T cells (% of CD3^+^CD4^+^ T cells, %)	0.616	0.004 *	−0.336	0.148	−0.309	0.185
ΔCD27^-^CD4^+^ T cells (% of CD3^+^CD4^+^ T cells, %)	−0.618	0.004 *	0.340	0.142	0.317	0.174

* *p* < 0.05. Abbreviations: ACT, asthma control test; FeNO, fractional exhaled nitric oxide; FEV_1_, forced expiratory volume in one second; FEV_1_%, forced expiratory volume % in one second; FGF, fibroblast growth factor; FVC, forced vital capacity; G-CSF, granulocyte-colony stimulating factor; IFN-γ, interferon-gamma; IL, interleukin; ILC, innate lymphoid cell; IP, interferon-γ inducible protein; MAIT, mucosal-associated invariant T; MCP, monocyte chemotactic protein; MIP, macrophage inflammatory protein; NK, natural killer; PDGF, platelet-derived growth factor; RANTES, regulated on activation normal T cell expressed and secreted; TARC, thymus and activation-regulated chemokine; Th, helper T; TNF-α, tumor necrosis factor-α; Treg, regulatory T; VAS, visual analog scale; VC, vital capacity.

## Data Availability

Not applicable.

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
