# Peer review of "Effect of Japanese Cedar Pollen Sublingual Immunotherapy on Asthma Patients with Seasonal Allergic Rhinitis Caused by Japanese Cedar Pollen"

_biomolecules, 2022, doi:10.3390/biom12040518_

Round 1

Reviewer 1 Report

  1. There is a lot of evidence that Japanese cedar pollen sublingual immunotherapy is safe and effective for the treatment of seasonal allergic rhinitis. This article further confirmed the therapeutic effect of Japanese cedar pollen sublingual immunotherapy on Japanese cedar pollen for bronchial asthma. It has moderate evidence and well clinical value.
  2. But the authors did not make  drugs score, which are very useful in determining the effectiveness of desensitization treatments. It is recommended that the author supplement this part of the content.

Author Response

We appreciate the reviewer's thoughtful review of our paper. We fully agree with the reviewer's kind comments. We have revised our manuscript accordingly. Responses to individual comments are given below. The reviewer's comments have resulted in substantial improvements in the manuscript.

Comments and Suggestions for Authors

  1. There is a lot of evidence that Japanese cedar pollen sublingual immunotherapy is safe and effective for the treatment of seasonal allergic rhinitis. This article further confirmed the therapeutic effect of Japanese cedar pollen sublingual immunotherapy on Japanese cedar pollen for bronchial asthma. It has moderate evidence and well clinical value.

Reply: We appreciate the reviewer's thoughtful review and comments.

  1. But the authors did not make drugs score, which are very useful in determining the effectiveness of desensitization treatments. It is recommended that the author supplement this part of the content.

Reply: We appreciate the reviewer’s important comments. We fully agree with your kind comments. Accordingly, we added medication scores and revised the manuscript. 

Revised sentences (page 3, lines 107-117): Subjective symptoms were evaluated using the asthma control test (ACT), the SACRA questionnaire, and medication score during the in-season (February to April) and off-season (June to October) before and after treatment. The contents of the ACT (Survey of Asthma Control in the Last 4 Weeks) and SACRA (Questionnaire and VAS Assessment of Rhinitis Symptoms and Asthma Symptoms) questionnaire have already been reported (14). Medication scores for allergic rhinitis and asthma are assigned to the different medica-tions as follows: 1 point = patient took nasal sprays and eye-drops; 2 points = patient took systemic antihistamines, inhaled corticosteroids, inhaled β2-agonists, inhaled anticholinergics, leukotriene receptor antagonist, suplatast tosilate, and theophylline; 3 points = patient took systemic corticosteroids, and omalizumab; and score for the use of each drug will be multiplied by two in case of use of maximum dose.

Revised sentences (page 6, lines 210-211): During the in-season, there was no difference in mean medication score between pre-SLIT and post-SLIT (7.2 ± 3.5 vs. 7.1 ± 3.1, P = 0.87).

Revised sentences (page 7, lines 240-241): During the off-season, there was no difference in mean medication score between pre-SLIT and post-SLIT (7.1 ± 2.5 vs. 6.1 ± 3.0, P = 0.14).

Reviewer 2 Report

The efficacy of sublingual immunotherapy (SLIT) in asthmatic patients with SAR-JCP and the dynamics 19 of the parameters before and after treatment in a real-world setting were investigated.

This topic is relevant and interesting in the field of allergy and SIT research.

The topic is original in the respect to local conditions and environmental factors (cedar pollen).

This manuscript adds new data about the immunological mechanism of SIT.

The manuscript is well written, clear and easy to read.

The conclusions are consistent with the evidence and arguments provided in the manuscript. They are addressed the main question.

Thus, my recommendation is "Accept".

Author Response

We appreciate the reviewer's thoughtful review and comments.

Reviewer 3 Report

The manuscript entitled "Effect of Japanese cedar pollen sublingual immunotherapy in asthma patients with seasonal allergic rhinitis caused by Japanese cedar pollen" done by Shoko Ueda et al. is very interesting in the field of SLIT. However, there are some minor issues which should be clarified by the authors:

  1. The study population was asthmatic people >20 years. Why was not >/= 18 years old for adult with asthma?
  2. The inclusion criteria were focused  only on asthmatic patients and not on both asthma and allergic rhinitis (AR). Please clarify it.
  3. The criteria to diagnosis of AR should be done in the Method section. Did the authors measure nasal FeNO? That is a good biomarker of AR.
  4. Did the authors realise skin prick test for study patients? If yes, please add it in the Results section.
  5. JCD is a special pollen due to its high molecular weight in compare to other airborn allergens. Please comment on it.

Author Response

We appreciate the reviewer's thoughtful review of our paper. We fully agree with the reviewer's kind comments. We have revised our manuscript accordingly. Responses to individual comments are given below. The reviewer's comments have resulted in substantial improvements in the manuscript.

Comments and Suggestions for Authors

 The manuscript entitled "Effect of Japanese cedar pollen sublingual immunotherapy in asthma patients with seasonal allergic rhinitis caused by Japanese cedar pollen" done by Shoko Ueda et al. is very interesting in the field of SLIT. However, there are some minor issues which should be clarified by the authors:

  1. The study population was asthmatic people >20 years. Why was not >/= 18 years old for adult with asthma?

Reply: We appreciate the reviewer’s important comments. Under the current Japanese law, adults are 20 years old or older, and 18 to 20 years old patients require their parental consent. Therefore, we recruited Japanese patients who are 20 years old or older in this study.

  1. The inclusion criteria were focused only on asthmatic patients and not on both asthma and allergic rhinitis (AR). Please clarify it.

Reply: We appreciate the reviewer’s important comments. We fully agree with your kind comments and apologize for the inaccuracy in this regard. Accordingly, we revised the manuscript to the inclusion criteria focused on both asthma and seasonal allergic rhinitis (SAR) caused by Japanese cedar pollen (JCP) (SAR-JCP).

Revised sentences (page 2, lines 86-93): The diagnosis of SAR-JCP was made by the criteria below, mainly based on rhinitis symptoms reported on the self-assessment of allergic rhinitis and asthma (SACRA) questionnaire and serum JCP-specific IgE antibodies as a reference to Japanese diagnostic criteria for allergic rhinitis: (1) in patients positive for ≧1 SACRA rhinitis symptom during the JCP dispersal season and JCP-specific IgE ≧ class 2, the diagnosis of SAR-JCP was confirmed; (2) in patients with no SACRA rhinitis symptoms during the JCP dispersal season or negative JCP-specific IgE (excluding patients taking antiallergic drugs), the diagnosis of SAR-JCP was excluded.

  1. The criteria to diagnosis of AR should be done in the Method section. Did the authors measure nasal FeNO? That is a good biomarker of AR.

Reply: We appreciate the reviewer’s comments. We fully agree with your kind comments but apologize that we cannot add the nasal FeNO measurements. As you pointed out, nasal FeNO is an important measurement to the diagnosis of AR, but it could not be done at our hospital. Unfortunately, nasal FeNO has not been measured.

  1. Did the authors realise skin prick test for study patients? If yes, please add it in the Results section.

Reply: We appreciate the reviewer’s comments. We fully agree with your kind comments but apologize that we cannot add the result of the skin prick test. As you pointed out, the skin prick test is an important measurement for diagnosing AR as well as nasal FeNO. However, we do not routinely perform the skin prick test for patients with asthma because the risk of anaphylaxis cannot be ruled out. Unfortunately, the skin prick test has not been performed in this study as it is used serum antigen-specific IgE with priority.

  1. JCD is a special pollen due to its high molecular weight in compare to other airborn allergens. Please comment on it.

Reply: We appreciate the reviewer’s comments. As you mentioned, the size of JCP is 30-38 μm, and JCP is a high molecular weight compared to other airborne allergens. However, it was recently found that a large number of fine particles (size 1 μm), called orbicules, are attached to the surface of the JCP. The orbicules can reach the respiratory tract and directly worsen asthma control. This issue has already been mentioned in lines 45-50 of the text.

Page 2 lines 45-55: JCP, which is one of the causative antigens of allergic rhinitis, has a large particle size of several tens of micrometers and does not reach the lower respiratory tract by itself. Therefore, it was previously thought to be unrelated to the exacerbation of asthma. However, it was recently found that a large number of fine particles, called orbicules, are attached to the surface of the JCP. Since the orbicules are approximately 1 μm, they can reach the airways and directly worsen asthma control. In Europe and North America, asthma has been reported to exacerbate frequently during the pollen dispersal season (7). In Japan, Hojo et al. reported that asthma symptoms worsen during the JCP dispersal season, even asthma associated with SAR-JCP (8). Respiratory dysfunction in SAR-JCP during JCP dispersal is thought to be caused by an increase in inflammatory mediators in the lower respiratory tract.